# Comparing machine learning and interpolation methods for loop-level calculations

**Ibrahim Chahrour⋆ and James D. Wells†**

Leinweber Center for Theoretical Physics, Physics Department,
University of Michigan, Ann Arbor, MI 48109-1040 USA

⋆ chahrour@umich.edu , † jwells@umich.edu

## Abstract

The need to approximate functions is ubiquitous in science, either due to empirical constraints or high computational cost of accessing the function. In high-energy physics, the precise computation of the scattering cross-section of a process requires the evaluation of computationally intensive integrals. A wide variety of methods in machine learning have been used to tackle this problem, but often the motivation of using one method over another is lacking. Comparing these methods is typically highly dependent on the problem at hand, so we specify to the case where we can evaluate the function a large number of times, after which quick and accurate evaluation can take place. We consider four interpolation and three machine learning techniques and compare their performance on three toy functions, the four-point scalar Passarino-Veltman $D_0$ function, and the two-loop self-energy master integral $M$. We find that in low dimensions ($d = 3$), traditional interpolation techniques like the Radial Basis Function perform very well, but in higher dimensions ($d = 5, 6, 9$) we find that multi-layer perceptrons (a.k.a neural networks) do not suffer as much from the curse of dimensionality and provide the fastest and most accurate predictions.



# List of Acronyms and Abbreviations

1. AE: Absolute Error

2. IDW: Inverse Distance Weighting

3. LGBM: Light Gradient Boosting Machine

4. MLP: Multi-layer Perceptron

5. MSE: Mean Squared Error

6. NN: Nearest Neighbors

7. RBF: Radial Basis Function

8. sAPE: Symmetric Absolute Percentage Error

9. SVGP: Stochastic Variational Gaussian Process

10. CV: Coefficient of Variation

# 1 Introduction

A pervasive problem in quantitative fields of study is the need to evaluate a function for which one has limited access to, either due to empirical constraints or high computational cost. The problem is worsened for high-dimensional settings where the input space is too large to be explored adequately with even relatively large amounts of data. This is often referred to as the *curse of dimensionality*. The standard way of dealing with this issue is to approximate the function either through interpolation[1] or regression. Examples of areas that require interpolation/regression include environmental sciences [1], astronomy [2], geology [3], and aerospace engineering [4].

The field of high-energy physics severely faces this issue, having to perform long and complicated computations before arriving at a precise prediction for an observable at colliders. The most common observables are arrived at by computing the cross-section of the scattering of two or more particles. To achieve suitable precision, calculations beyond the leading order (LO) are needed [5]. Beyond LO, one encounters a large number of loop diagrams which need to be reduced to a set of master integrals, the evaluation of which can take hours on a modern quad-core CPU for a single phase space point [6]. Moreover, if the scattering is between partons inside hadrons, then one needs to know the parton distribution functions (PDFs) that give the probability density of finding a parton with some momentum fraction of the parent hadron. These PDFs must be fit from experimental data as they cannot be derived from perturbation theory, which once again raises the question of how to choose between function approximation techniques.

Recently, machine learning (ML) techniques have been applied to a variety of situations in high-energy physics. In [7], normalizing flows [8] were used to reduce the uncertainties in the evaluation of multi-loop integrals, leading to a speedup in their precise numerical computation. On the regression front, many attempts have been made to approximate matrix elements and cross-sections of processes in the Standard Model and theories beyond it. In [9], neural networks were used to approximate the cross-section of chargino production in the phenomenological Minimally Supersymmetric Standard Model (pMSSM-19) achieving low relative errors with a prediction speedup on the order of $\mathcal{O}(10^6)$ (excluding training time) over the full calculation. The authors in [10] utilized `XGBoost` [11], a gradient boosting machine, to approximate the matrix element of gluon fusion to Z bosons, achieving a speedup of a factor of 20 compared with the full calculation (assuming a training time of 7 minutes as stated in [10]). In [12–14], neural networks were used to approximate high-dimensional squared matrix elements of jet production from $e^+e^-$ annihilation and diphoton production. There, an ensemble of neural networks is trained to quantify the uncertainty of the prediction, as well as improve the performance. Counting the training and prediction time, a speedup of a factor of 10 compared to the full calculation was achieved in [12]. On the other hand, the authors in [15] leverage distributed Gaussian process (GP) regression to approximate cross-sections in the MSSM. Similarly, the authors in [16] used GPTree [17], an interpolant based on GP and KDTrees, to interpolate the two-loop amplitude of $q\bar{q} \to \gamma\gamma\gamma$ achieving faster evaluation and an uncertainty within those of a Monte Carlo approach [18]. The NNPDF collaboration [19] use neural networks as unbiased interpolants of PDFs for hadronic studies at colliders.

The success of the previous methods have made them standard techniques in the field of high-energy physics. Still, it is interesting to ask whether there are general principles that can guide the selection of the approximation method and whether machine learning is really necessary, or do traditional interpolation techniques suffice. Furthermore, determining the best method can be highly dependent on the task at hand. Many papers have looked at comparing

---

[1]We use the term interpolation in its original sense, where the interpolating function attains the function value exactly at some known data.

various methods for their own fields and problems. In [20], four interpolation techniques were compared on 2-dimensional spatial data with mixed results as to the best interpolant. Similarly in [21], three interpolation techniques were compared on a low number of electric field magnitude data in two dimensions, with inverse distance weighting performing the best. On the other hand, a novel variation of support vector regression was compared to interpolation techniques on seismic data in low dimensions with hundreds of thousands of data points [22], leading to competitive performance. A number of interpolation and machine learning models were put to the test on various materials science datasets in [23]. There, the datasets were high-dimensional but sparse, with GP regression performing the best. In the field of ecology [24], the distribution of the data was found to have a large effect on the performance of the interpolation/regression techniques, with some being affected more than others. We can see from these studies that it can be difficult to make general statements about approximation methods, even in low dimensions. Thus, it is crucial to specify our situation in this study and describe the particular problem we would like to solve.

We aim at approximating computationally expensive functions appearing in the calculation of scattering cross-sections in high-energy physics. These functions can be low- or high-dimensional, depending on the process at hand. But, we assume that we can evaluate these functions a large number of times, after which quick and accurate evaluation can be performed. To that end, we assess four interpolation and three machine learning techniques in various dimensions, comparing their accuracy and evaluation speed. Throughout the paper, we will refer to the interpolation techniques as interpolants, the regression techniques as models, and the combination as approximants.

This paper is organized as follows: first we present the interpolation methods in Sec. 2, describing briefly how each method works. Then we present the ML regression methods in Sec. 3, highlighting the chosen settings and hyperparameters. After that, we list the test functions in Sec. 4 and describe their output distribution through data statistics. For model selection and method comparison, we explain the cross-validation procedure in Sec. 5 and list the metrics used to compare the approximants. We then present the results in Sec. 6 and discuss the performance of the approximants, highlighting the features and drawbacks of each. Lastly, we present new directions for this research in Sec. 7.

# 2 Interpolation Methods

We start by presenting the interpolation methods, where the interpolating function passes through all known data points.

## 2.1 Nearest-Neighbor Interpolation (NN)

Perhaps the simplest interpolation technique, the method of nearest-neighbors assigns the value of the new point to that of the nearest known data point. As such, the resulting interpolant is piece-wise constant. This technique is commonly used in image interpolation because of its low computational complexity [25]. We can write the interpolant as

$$\hat{f}_{\text{NN}} = f\left(\underset{i}{\arg\min} \, ||\mathbf{x} - \mathbf{x}_i||_2\right), \tag{1}$$

where $\mathbf{x}$ is the unknown point, $\mathbf{x}_i$ are the known points, $|| \, ||_2$ is the Euclidean distance, and $f$ returns the known values. The advantage of NN interpolation is its simplicity, serving as a lookup table for unknown points.

In this analysis, we use SciPy's [26] `NearestNDInterpolator` to perform the interpolation which utilizes the KD Tree algorithm [27] to find the nearest neighbors.

## 2.2 Linear Interpolation on Regular Grid (Grid)

Initially, we considered linear interpolation on irregular data which proceeds first by triangulating the input data points (typically Delaunay triangulation [28]). Then, interpolation on a new data point is performed by checking which $d$-simplex the new point falls within and computing the weighted average of the vertices of the $d$-simplex. The weights are given by the barycentric coordinates of the new point with respect to its host simplex. However, applying the triangulation on a large dataset and storing it required gigabytes of data, forcing us to consider interpolation on a regular grid instead. This is clearly less flexible, but nonetheless, it is worth pursuing to check whether the performance trumps the demands of flexibility.

The task here is simplified because no triangulation is required. Given $N$ known data points on a grid in $d$ dimensions, an unknown point will lie in a hyperrectangle whose $2^d$ vertices are known points. The interpolation proceeds by partitioning the hyperrectangle into $2^d$ hypervolumes sharing the unknown point as a common vertex. The assigned value is then the sum of the known $2^d$ points weighted by the normalized diagonally opposite hypervolumes [29]. The normalization here is by the volume of the entire hyperrectangle.

In this analysis, we use SciPy's [26] `RegularGridInterpolator` to perform the interpolation.

## 2.3 Inverse Distance Weighting (IDW)

Introduced by Shepard in 1968 [30], IDW has become widely used in geographic information systems (GIS) for its ease of use and interpretability. The complexity of interpolating a point is $\mathcal{O}(N)$ which is desirable for our case where $N$ is large. To interpolate a new point, IDW takes the weighted average of all the known points with the weights corresponding the inverse of the distance between the new and known points. Thus we can write

$$\hat{f}_{\text{IDW}}(\mathbf{x}) = \frac{\sum_{i=1}^{N} w_i(\mathbf{x}) y_i}{\sum_{i=1}^{N} w_i(\mathbf{x})}, \tag{2}$$

where $N$ is the number of known points, $y_i$ is the value of the ith known point, and $w_i$ is given by

$$w_i(\mathbf{x}) = \frac{1}{||\mathbf{x} - \mathbf{x}_i||_2^p}. \tag{3}$$

We choose $p = 1$ in this analysis. The interpolation is performed using Photutil's [31] `ShepardIDWInterpolator`.

## 2.4 Radial Basis Function (RBF) Interpolation

Originally developed by Hardy [32] and later shown to consistently outperform its competitors by Franke [33], RBF has achieved immense popularity as an interpolation technique. The idea of RBF is to write our approximation as a linear combination of a function that depends only on the distance between two points (hence, radial). At each known point, there is an RBF centered at that point which gives us the interpolant:

$$\hat{f}_{\text{RBF}}(\mathbf{x}) = \sum_{i=0}^{N-1} w_i \varphi(||\mathbf{x} - \mathbf{x}_i||_2), \tag{4}$$

where $N$ is the number of known data points, $\mathbf{x}$ is the point we are interpolating, $\mathbf{x}_i$ are the known points, $\varphi$ is the radial basis function, and $w_i$ are the weights. To fix the weights, we require that our interpolating function pass through all known points, which gives us $N$ linear equations in the weights. The complexity of solving this system is $\mathcal{O}(N^3)$ which can be

prohibitive for large amounts of data. Furthermore, the memory requirements of interpolating at a given point is $\mathcal{O}(N^2)$ [34]. To overcome this, we limit the interpolant to the k-nearest neighbors rather than the entire dataset at prediction time. In particular, we choose 150-nearest neighbors based on a speed versus performance trade-off (see appendix B). We note that there are other techniques of speeding up RBF that are worth exploring, such as the compactly supported radial basis functions (CSRBFs) with spatial subdivision [34,35].

In this analysis, we use SciPy's [26] `RBFInterpolator` with the default *thin-plate spline* RBF.

## 3 Regression Methods

Unlike interpolation, regression does not require the approximating function to assume the values of the known points. A regression model typically proceeds by minimizing some objective function (e.g. mean-squared error). Finding the optimal model parameters that minimize the objective is a difficult task requiring first- or second-order gradient methods. The following regression models all fall under the umbrella of machine learning.

### 3.1 Multilayer Perceptron (`MLP`)

Multilayer Perceptrons[2] are used for a variety of purposes such as classification, regression, and variational inference. It has been shown that MLPs with a single hidden layer, sufficiently many hidden nodes, and suitable activation function can approximate any continuous function on a compact set in $\mathbb{R}^n$ to arbitrary accuracy [36, 37]. This provides great motivation for the use of MLPs in regression tasks. An MLP can be viewed as a function $\mathcal{N}(\mathbf{x}; \boldsymbol{\theta})$ where the dimension of the input $\mathbf{x}$ determines the number of nodes in the input layer, and the parameters $\boldsymbol{\theta}$ are adjusted during the training process to minimize the objective. We use a fully-connected architecture with each layer of the form

$$l(\mathbf{a}) = \phi\left(W \cdot \mathbf{a} + \mathbf{b}\right),\tag{5}$$

where $\mathbf{a}$ is the input from the previous layer, $W$ is the $h \times k$ weight matrix of two connected layers, $\mathbf{b}$ are the biases of the layer, and $\phi$ is a nonlinear function called the activation. The dimensions $h$ and $k$ represent the number of nodes in the previous and current layer respectively.

The construction of MLPs is performed using `Tensorflow` [38] and `Keras` [39]. Performing hyperparameter optimization is computationally expensive so we rely on empirical tests to guide the settings. We use an architecture of 8 hidden layers with 64 nodes each, applying the Gaussian Error Linear Unit (GELU) [40] activation and LSUV weight initialization [41]. Using the Adam [42] optimizer, we minimize either the mean squared error (MSE) loss

$$L_{\text{MSE}} = \frac{1}{n} \sum_{i=1}^{n} (\mathcal{N}(\mathbf{x}_i) - y_i)^2\tag{6}$$

or the mean absolute percentage error (MAPE)

$$L_{\text{MAPE}} = \frac{1}{n} \sum_{i=1}^{n} \left| \frac{\mathcal{N}(\mathbf{x}_i) - y_i}{y_i} \right|,\tag{7}$$

---

[2]Also known as deep or artificial neural networks. We choose the name multilayer perceptron to avoid confusion with nearest neighbors when abbreviating.

where $n$ is the batch size and $y_i$'s are the true values. In general, we use the MSE loss except on the $D_0$ function introduced in Sec. 4. We train on an NVIDIA Tesla V100 GPU for a maximum of 4000 epochs with the EarlyStopping callback that stops training when no improvement has been made over 400 epochs. The batch size is set to either 1000 or 5000 training points. The learning rate is set to the default value 0.001 and is decayed every 100 epochs by a factor of 0.85.

## 3.2 Light Gradient Boosting Machine (`LGBM`)

The idea of boosting is to combine many weak learners into one strong model. `LGBM` [43] is a decision tree model that is lightweight and fast, achieving similar performance to `XGBoost` [11] while consuming much less time and memory [43]. `LGBM` uses leaf-wise growth rather than the typical level-wise growth [44] when building the tree. Furthermore, typical decision trees use pre-sorted algorithms [45] to find the best split locations, whereas `LGBM` uses histogram-based algorithms which speed up training and require less memory [43].

Like other machine learning models, `LGBM` comes with many hyperparameters that need to be tuned. Similar to the MLP case, we rely on empirical tests to guide the settings. We highlight what we deem to be the most important hyperparameters and the chosen values:

- `num_leaves` = 500

- `max_depth` = 10

- `max_bin` = 300

- `num_iterations` = 1500

- `sub_samples` or `bagging_fraction` = 0.5

In determining the booster, we tried the default `gbtree` and `dart` [46], which applies the idea of dropout in deep learning [47] to boosted trees to avoid over-fitting. We find that `dart` generalizes much better than `gbtree` so we choose `dart` as our booster. The implementation of `LGBM` is done through the scikit-learn API [48].

## 3.3 Stochastic Variational Gaussian Process (`SVGP`)

The textbook definition of a Gaussian process (GP) is that it "is a collection of random variables, any finite number of which have a joint Gaussian distribution" [49]. For a given mean function $m(\mathbf{x})$ and covariance function $k(\mathbf{x}, \mathbf{x}')$, a GP is completely specified [49] and can be denoted

$$f_{\text{GP}}(\mathbf{x}) \sim \mathcal{GP}\left(m(\mathbf{x}), k\left(\mathbf{x}, \mathbf{x}'\right)\right). \tag{8}$$

A common choice for the kernel, which we use here, is the squared exponential function:

$$k(r) = \alpha^2 \exp\left\{-\frac{r^2}{2l}\right\}, \tag{9}$$

where $r$ is the Euclidean distance between $\mathbf{x}$ and $\mathbf{x}'$, $l$ is the lengthscale parameter, and $\alpha$ is the variance parameter. The kernel is promoted to a matrix $\mathbf{K}$ when dealing with a training dataset $\mathcal{D} = (\mathbf{X}, \mathbf{y})$ of $N$ observations, and the optimization of the parameters $l$ and $\alpha$ can be achieved through the maximization of the marginal likelihood [50]

$$\log p(\mathbf{y}) = -\frac{N}{2}\log 2\pi - \frac{1}{2}\log|\mathbf{K}| - \frac{1}{2}\mathbf{y}^\top (\mathbf{K})^{-1}\mathbf{y}. \tag{10}$$

Given an unknown point $\mathbf{x}_*$, the mean and variance of the Gaussian predictive distribution are given by [50]

$$
\begin{aligned}
\mu_{\text{GP}}(\mathbf{x}_*) &= \mathbf{k}_*(\mathbf{K})^{-1}\mathbf{y}, \\
\sigma^2_{\text{GP}}(\mathbf{x}_*) &= k_{**} - \mathbf{k}_*(\mathbf{K})^{-1}\mathbf{k}_*,
\end{aligned}
\tag{11}
$$

where $\mathbf{k}_* = k(\mathbf{x}_*, \mathbf{X})$ and $k_{**} = k(\mathbf{x}_*, \mathbf{x}_*)$. A nice feature of using GP's is that we can obtain an uncertainty $\sigma_{\text{GP}}$ for our prediction $\mu_{\text{GP}}$ without extra work. However, computing $\mu_{\text{GP}}$ and $\sigma_{\text{GP}}$ involves the inversion of the $N \times N$ kernel covariance matrix $\mathbf{K}$ which exhibits a time complexity $\mathcal{O}(N^3)$ that becomes prohibitive for large $N$. The idea of SVGP [51] is to use a subset of the training data called *inducing variables* that aim at summarizing the data, simultaneously approximating the posterior $p(f_{\text{GP}}|\mathbf{y})$ through variational inference [52]. The final time complexity for $m$ inducing variables is $\mathcal{O}(m^3)$. This allows us to perform GP regression on very large datasets.

We use the `GPFlow` [53] package to fit the SVGP model. The number of inducing variables is set to 2000 and the optimization of the evidence lower bound (ELBO) is done through Adam [42]. We select a batch size of 5000 points and optimize for 100,000 epochs on a GPU. The learning rate is decayed every 1,000 epochs by a factor of 0.99.

## 4 Test Functions

The goal of this analysis is to approximate 'black-box' functions, i.e. functions that can only be evaluated without much knowledge of their internal structure. However, here we consider known functions to understand the performance of each technique and to facilitate the comparison. We look at multidimensional polynomial, Camel, and periodic functions. The polynomial and Camel functions were used as tests for the `i-flow` [54] integration code. In addition, we look at two integral functions that appear widely in loop-level calculations in the Standard Model. The first is the four-point scalar Passarino-Veltman integral function [55], and the second is the two-loop self-energy master integral.

In all of the following, we uniformly sample the unit hypercube $[0,1]^d$.

### 4.1 Toy Functions

The three toy functions are given by

$$
f_{\text{poly}}(\mathbf{x}) = \sum_{i=1}^{d} -x_i^2 + x_i,
\tag{12}
$$

$$
f_{\text{Camel}}(\mathbf{x}) = \frac{1}{2(\sigma\sqrt{\pi})^d}\left(\exp\left(-\frac{\sum_i\left(x_i - \frac{1}{3}\right)^2}{\sigma^2}\right) + \exp\left(-\frac{\sum_i\left(x_i - \frac{2}{3}\right)^2}{\sigma^2}\right)\right),
\tag{13}
$$

$$
f_{\text{periodic}}(\mathbf{x}) = \bar{x}\prod_{i=1}^{d}\sin 2\pi x_i,
\tag{14}
$$

where $d$ is the dimension, $\bar{x}$ is the mean of $\mathbf{x}$, and as in [54] we choose $\sigma = 0.2$. The polynomial function has no 'difficult' features to learn like sharp peaks or oscillations, so it should be the simplest to approximate. The multidimensional Camel is chosen because it contains two peaks that get harder and harder to locate at large dimensions. The sharpness of the peaks is controlled by the width $\sigma$. The periodic function is chosen to test the approximation techniques on oscillating functions. As we increase the dimension, the number of peaks and valleys rises quickly making this function very difficult in high dimensions.

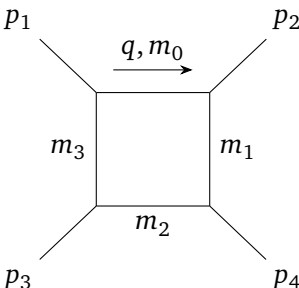

Figure 1: Feynman diagram of the Scalar Passarino-Veltman $D_0$ function.

A common technique which we employ here is to transform the outputs of the function before fitting/training. In this section we merely state the transformations and in Sec. 4.3 we show the consequences of such transformations. For the Camel function, we perform a simple log scaling of the outputs since all values are positive and the range of the function is quite high. For the periodic function, we perform a signed cube root scaling given by:

$$\text{sgn}(y) \cdot \sqrt[3]{|y|}, \tag{15}$$

where sgn is the sign function. The polynomial outputs are not scaled. In general, we fit/train on both scaled and unscaled outputs and report the best performance.

## 4.2 Loop Integral Functions

To perform high precision theoretical calculations in particle physics, one must compute the contribution of loop diagrams appearing in the Feynman diagram expansion of a scattering process. These loop diagrams give rise to Feynman integrals whose numerical calculation is very expensive, especially at higher loop order. We consider two Feynman integrals to approximate: the scalar four-point Passarino-Veltman function at one-loop, and the two-loop self-energy master integral.

### 4.2.1 Scalar Passarino-Veltman $D_0$

At the one-loop level, it has been shown [55] that any tensor integral can be reduced to a set of 1-, 2-, 3-, and 4-point scalar integrals, the last of these being the most complicated. This 4-point scalar integral is typically referred to as the Passarino-Veltman $D_0$ function.

The function depends on the kinematics of the external particles (all momenta are incoming) and the masses of the internal particles and is given by:

$$D_0\left(s_1, s_2, s_3, s_4, s_{12}, s_{23}, m_0, m_1, m_2, m_3\right) = \tag{16}$$

$$C_0 \int \mathrm{d}^r q \frac{1}{\left(q^2 - m_0^2\right)\left[(q+p_2)^2 - m_1^2\right]\left[(q+p_2+p_4)^2 - m_2^2\right]\left[(q+p_2+p_3+p_4)^2 - m_3^2\right]},$$

where $q$ is the loop momentum, $s_{ij} = (p_i + p_j)^2$, $s_i = p_i^2$, and $C_0$ is a constant from dimensional regularization. Although not very expensive to evaluate, $D_0$ serves as a good starting point in determining how hopeful our project should be moving to higher loops.

By fixing certain quantities, we can look at this function in various dimensions up to $d = 9$. Although there are 10 independent variables, one can rewrite $D_0$ in the following way for any of the inputs:

$$D_0\left(x_1, x_2, x_3, x_4, x_5, x_6, x_7, x_8, x_9, x_{10}\right) = \frac{1}{x_5} D_0\left(y_1, y_2, y_3, y_4, 1, y_6, y_7, y_8, y_9, y_{10}\right), \tag{17}$$

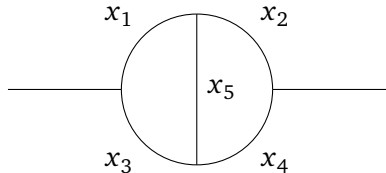

Figure 2: Two-loop self-energy diagram corresponding to the function $M$.

where $y_i = x_i/x_5$ effectively reducing the maximum dimension to 9. We select somewhat arbitrary regions of the parameter space to arrive at the three $D_0$ functions in $d = 3, 6$, and 9 dimensions:

$$D_0^{(3)} = \text{Re}\left[D_0\left(0.01, 0.04, 0.16, \frac{x_4}{4}, 1, u(x_6), \frac{x_7}{2}, \frac{x_7}{2}, \frac{x_7}{2}, 0.2\right)\right], \tag{18}$$

$$D_0^{(6)} = \text{Re}\left[D_0\left(0.01, 0.04, 0.16, \frac{x_4}{4}, 1, u(x_6), x_7, x_8, x_9, x_{10}\right)\right], \tag{19}$$

$$D_0^{(9)} = \text{Re}\left[D_0\left(\frac{x_1}{4}, \frac{x_2}{4}, \frac{x_3}{4}, \frac{x_4}{4}, 1, u(x_6), \frac{x_7}{2}, \frac{x_8}{2}, \frac{x_9}{2}, \frac{x_{10}}{2}\right)\right], \tag{20}$$

where $u(x_6)$ is a function given in appendix A. These three functions will also be referred to as D3, D6, and D9 respectively. It is important to note that although the previous toy functions were differentiable everywhere, $\text{Re}[D_0]$ is not and has non-trivial analytic structure. The evaluation of $D_0$ is performed using *Package-X* [56], which is interfaced with the *COLLIER* library [57–60] through the *CollierLink* interface. The outputs are scaled by equation 15 if better performance is found.

### 4.2.2 Two-loop Self-energy Master Integral $M$

Theories beyond the Standard Model often posit fields that affect the physical mass of known particles by contributing to the self-energy diagrams [61]. As such, it is important to have precise theoretical predictions at the two-loop level for the self-energy corrections. Similar to the one-loop case, one can reduce two-loop self-energy diagrams to a set of basis integrals [62]. The master integral $M$ corresponding to the topology in Fig. 2 is free of divergences and will be our fifth test function.

The form of $M$ is given by [61]

$$M(x_1, x_2, x_3, x_4, x_5) = \tag{21}$$
$$\lim_{\epsilon \to 0} C^2 \int d^r k\, d^r q\, \frac{1}{[k^2 + x_2][q^2 + x_2][(k-p)^2 + x_3][(q-p)^2 + x_4][(k-q)^2 + x_5]},$$

where $x_1$, $x_2$, $x_3$, $x_4$, and $x_5$ are the masses of the internal particles, and $C$ is an overall constant appearing in dimensional regularization. Again, we only take the real part: $\text{Re}[M]$, arriving at a single-valued 5 dimensional function. The evaluation of the master integral is performed through the *TSIL* program [61]. We do not scale the outputs of this function.

### 4.3 Characterizing the Distributions of the Test Functions

Now that we have listed the test functions, we would like to characterize how difficult it would be to approximate them based on their distributions. A common method [24, 63] is to consider data variation described by the coefficient of variation (CV) and the moments of the

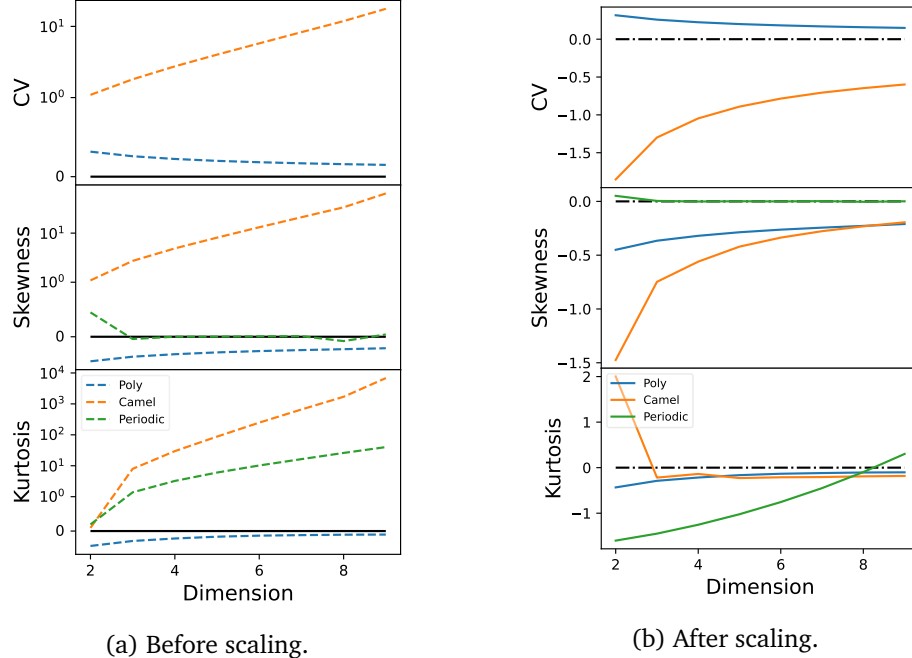

(a) Before scaling.

(b) After scaling.

Figure 3: Three data statistics: coefficient of variation, skewness, and kurtosis in various dimensions for the polynomial, Camel, and periodic functions.

distributions[3]. These statistics are given by:

$$\text{CV} = \frac{\mu}{s}, \tag{22}$$

$$S = \frac{\sum_{i=1}^{N}(y_i - \bar{y})^3/N}{s^3}, \tag{23}$$

$$K = \frac{\sum_{i=1}^{N}(y_i - \bar{y})^4/N}{s^4} - 3, \tag{24}$$

where $\mu$ is the mean, $s$ is the standard deviation, $S$ is the Fisher-Pearson coefficient of skewness, and $K$ is the coefficient of excess kurtosis [64]. In general, non-zero skewness indicates an asymmetry in the distribution, leaning to the left or right. On the other hand, kurtosis measures the degree to which a distribution contains outliers [65]. Fig. 3 shows the three statistics CV, skewness, and kurtosis in increasing dimensions for the polynomial, Camel, and periodic functions (a) before and (b) after scaling.

The polynomial function exhibits low CV, skewness, and kurtosis regardless of dimension, so we do not scale the true values. On the other hand, the Camel function exhibits increasingly large CV, skewness, and kurtosis as the dimension increases (Fig. 3a). This motivates a log-scaling of the true values prior to fitting/training, which dramatically reduces all three statistics and flips the sign of $K$ (Fig. 3b). The periodic function has a mean around zero so its CV is very large and not plotted here. Prior to cube root scaling, its skewness is nearly 0 and remains so after scaling, indicating a symmetric distribution that is preserved with scaling. Lastly, the kurtosis of the periodic function is positive and increases with dimension to a maximum of about 40 in 9 dimensions (Fig. 3a), whereas after cube root scaling, it turns negative until 8 dimensions where it attains a much smaller positive kurtosis of 0.3 in 9 dimensions. Turning to the loop integral functions, table 1 lists the values of CV, skewness, and kurtosis for the $D_0$ functions before and after scaling, and those for $M$ which is not scaled. Again, we see the the

---

[3]We only consider the third and fourth moments.

Table 1: Data statistics before and after scaling for the D3, D6, D9, and $M$ functions.

| Function | CV | Skewness | Kurtosis |
|---|---|---|---|
| D3 | Before scaling: $-2.69$ | $-0.9$ | $79.0$ |
| | After scaling: $-2.46$ | $0.46$ | $-1.12$ |
| D6 | $25.0$ | $-12.5$ | $2526$ |
| | $2.48$ | $-1.06$ | $1.38$ |
| D9 | $-4.83$ | $-60.0$ | $9053$ |
| | $-3.04$ | $0.08$ | $-0.88$ |
| $M$ | $0.57$ | $1.64$ | $9.57$ |

benefits of scaling the outputs in reducing the absolute values of skewness and kurtosis, while also flipping the sign of kurtosis for D3 and D9.

## 5 Cross-validation and Evaluation Metrics

Our assumption of working with a costly black-box function permits us to evaluate it $N$ times. For regression, we split our generated data into training, validation, and testing sets denoted by $N_{\text{train}}$, $N_{\text{val}}$, $N_{\text{test}}$ respectively. We start with the training phase where the model is fed the training data in batches and the parameters are adjusted to minimize the objective function. During training, we use the validation data to monitor the performance of the model on data it hasn't been trained on. This allows us to stop training when our models start to overfit. Finally, the testing phase is when we apply our model to unseen data to measure the performance "in the real world". For interpolation, we do not need a validation set, so the interpolation data is just $N_{\text{train}} + N_{\text{val}}$. We use a common testing dataset on all approximants. For all functions, we choose $N_{\text{test}} = 1\text{M}$ and $N_{\text{val}} = 10\%N_{\text{train}}$. The training data is either 4M for the two-loop self-energy function $M$ or 5M for the remaining functions.

To evaluate the models and compare them, we look at the distribution of absolute errors (AE) and symmetric absolute percent errors (sAPE):

$$\text{AE}_i = |y_i - \hat{f}_i|, \tag{25}$$

$$\text{sAPE}_i = \frac{2 \cdot |y_i - \hat{f}_i|}{|y_i| + |\hat{f}_i|} \cdot 100\%, \tag{26}$$

where $y_i$ is the true value and $\hat{f}(\mathbf{x}_i)$ is the prediction of the approximant. The symmetric version of the percent errors is chosen since our functions can vanish at some inputs which leaves the usual percent error undefined. These two metrics will be presented through boxplots (see Fig. 4 and caption for details) where outliers are suppressed for plotting purposes. The mean values will be shown as green markers on the boxplots in Sec. 6. We also look at the coefficient of determination, $R^2$, given by

$$R^2 = 1 - \frac{\sum_i (y_i - \hat{f}_i)^2}{\sum_i (y_i - \bar{y})^2}, \tag{27}$$

where $\bar{y}$ is the mean of the true values. This coefficient cannot exceed 1 (the case where the predicted values perfectly coincide with the true values), but can be negative which indicates worse performance than a baseline of the average of the function. It is also scale-free, which is useful in comparing performance across dimensions. In addition to boxplots that quantitatively summarize the performance of each approximant, it is useful to look at prediction versus truth

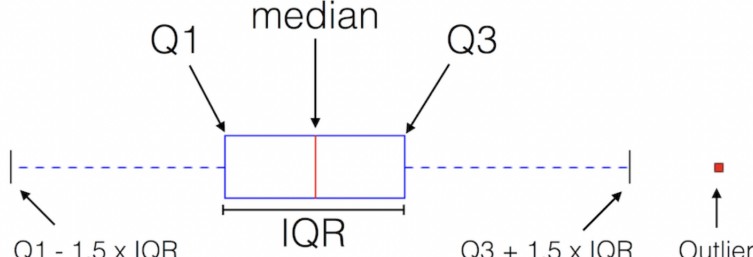

Figure 4: Boxplot illustration from Sebastian Raschka [66]. The image has been cropped to remove explanatory text. Q1 denotes the first quartile i.e. the value below which is 25% of the data points. The median or Q2 divides the data points in half. Q3 indicates the values below which is 75% of the data points. The interquartile range IQR is the range between Q1 and Q3. Values that lie below Q1−1.5×IQR and above Q1+1.5×IQR are plotted separately as points and are referred to as outliers.

plots which can reveal the scales at which the approximants are performing better or worse. We show these plots in reference [67]. We also consider secondary metrics such as the prediction time and disk size that are also of interest for speed and distribution.

# 6 Results

We analyze the performance of the approximants in 3, 6, and 9 dimensions on the toy functions and $D_0$. We give a separate analysis for the 5-dimensional function $M$. The `Grid` method is not applied to the loop integral functions since the data was generated randomly.

## 6.1 3 Dimensions

We begin with results in 3 dimensions. Figure 5 shows boxplots of sAPEs in 3 dimensions for the three toy functions and D3. The red colors indicate interpolation while blue colors indicate regression methods, and the green markers indicate the mean value. For all four functions, RBF achieves the lowest median and Q3 sAPE values. MLPs and `Grid` consistently achieve lower sAPEs than the remaining methods. Although RBF achieves the lowest medians, it results in larger means for the periodic and D3 functions where `Grid` and MLP achieve the lowest mean sAPEs respectively. This indicates that RBF produces many outliers.

In addition to percent differences, often we are interested in absolute differences which offer a better sense of scale. In Fig. 6, we see that RBF achieves the lowest AEs, both in terms of median and mean values, for all functions except D3 where MLP achieves a lower mean AE. This indicates that MLP approximates large values better than RBF, probably due to cube root scaling, while RBF is much more accurate overall. This can also explain the higher $R^2$ value of MLP in Fig. 7 compared to RBF. Worth noting is that SVGP performs poorly and cannot find a good fit for D3 compared to the other approximants, likely due to the inducing variables not being able to summarize its high variability.

## 6.2 6 Dimensions

In 6 dimensions, the story begins to change. Figure 8 shows boxplots of sAPEs in 6 dimensions for the three toy functions and D6. For all four functions, MLP achieves the lowest sAPE values both in terms of median and mean values. Aside from the performance of MLP, there is no

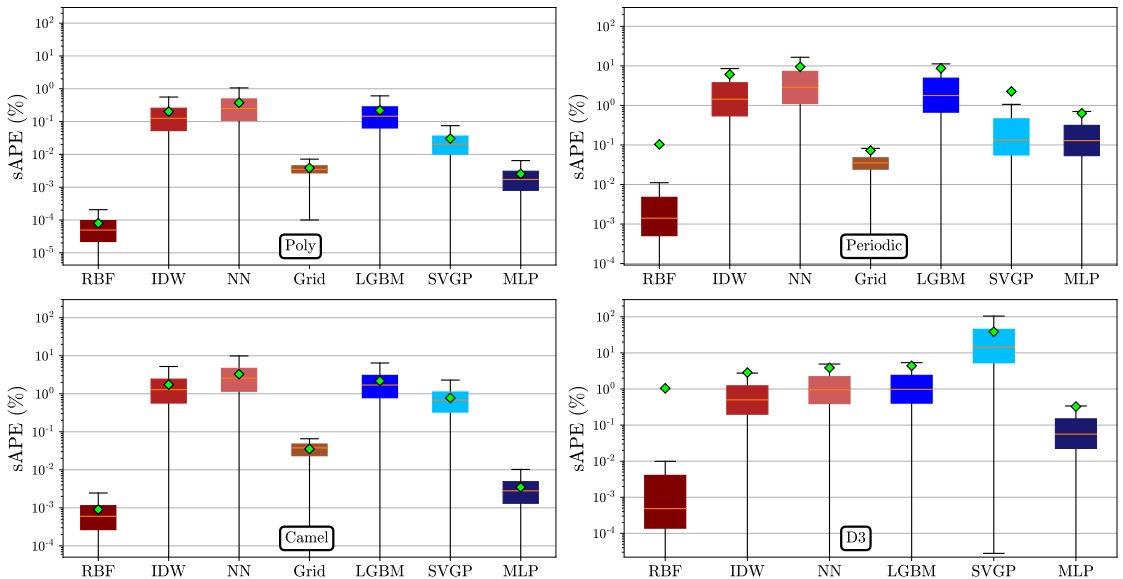

Figure 5: Boxplots of the symmetric absolute percent error (%) for four test functions in 3 dimensions. Red colors indicate interpolation while blue colors indicate regression methods.

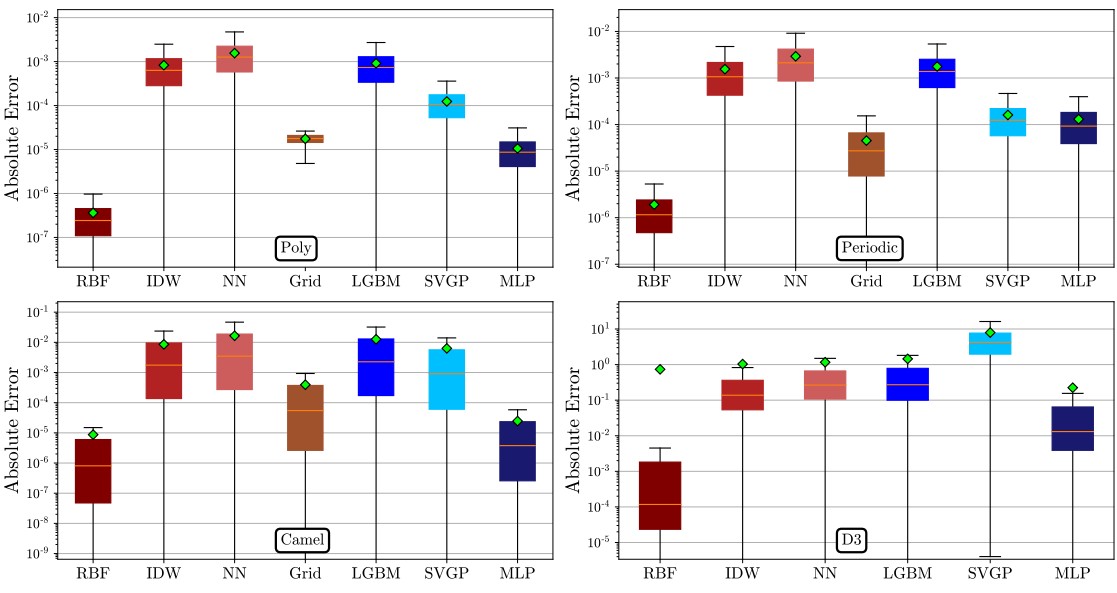

Figure 6: Boxplots of the absolute errors for four test functions in 3 dimensions. Red colors indicate interpolation while blue colors indicate regression methods.

consistent behavior when comparing the remaining methods. RBF achieves the 2nd lowest median sAPEs except on the polynomial where SVGP performs very well but does poorly on the remaining functions. NN does the worst on the toy functions, but better than LGBM and SVGP on D6. The upshot is that already in 6 dimensions, we see MLP consistently performing better than other approximants. This statement is aided by the AE plots (Fig. 9) and the $R^2$ plots (Fig. 10), which show the lowest AEs and highest $R^2$ values for MLP.

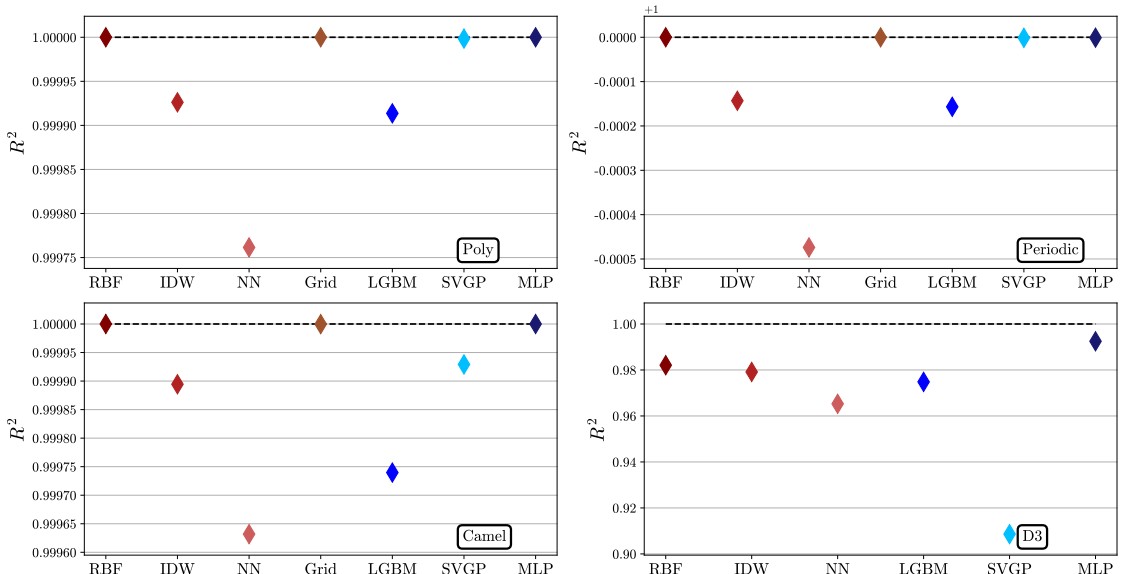

Figure 7: $R^2$ values in 3 dimensions for the three toy functions and D3.

## 6.3  9 Dimensions

In 9 dimensions, we see the superiority of MLP compared to the other approximants. Figure 11 shows boxplots of sAPEs in 9 dimensions for the three toy functions and D9. Once again, for all four functions, MLP achieves the lowest sAPE values both in terms of median and mean values. For the polynomial, the relative performance of the approximants is nearly identical to that in 6 dimensions. This confirms the low variability of this function established in Sec. 4.3. For the periodic function, all approximants except MLP perform very poorly with very high sAPEs. We find that cube root scaling is essential for good MLP performance, otherwise training is stuck at a local minimum. However, even with cube root scaling for the other approximants, we find no significant difference in performance. In particular, SVGP predicts all values to be identically near 0, again due to the inducing variables' inability to capture the periodic nature of the function in such high dimensions. For the Camel function, RBF outperforms the remaining approximants, with Grid having the highest sAPEs. SVGP produces a few extremely large outliers (Fig. 12) which makes it unreliable as an approximant in this case. We find that log scaling produces much better results than fitting/training on the raw data. Lastly for the D9 function, interestingly, we find that IDW has the 2nd lowest median and mean sAPEs, whereas in lower dimensions, it consistently had higher sAPEs than RBF. Looking at the $R^2$ values for D9 in Fig. 13, we see that RBF, NN, and LGBM have negative $R^2$, performing worse than a function that returns the mean of D9 everywhere. Putting it all together, it is clear that MLP is superior to the other approximants in 9 dimensions.

## 6.4  Two-loop Master Integral (5 Dimensions)

The last function we consider is the 5-dimensional two-loop self-energy master integral $M$. Figure 14 shows the AEs, sAPEs, and $R^2$ values of the approximants. Here again, MLP achieves the lowest median AE and sAPE (also the lowest mean AE and sAPE), as well as the highest $R^2$ value. It also has a much smaller IQR as opposed to RBF which has a wide IQR. Our previous results for RBF in 3 dimensions (see Sec. 6.1) showed great performance for all functions, but already in 5 dimensions we see MLP outperforming it.

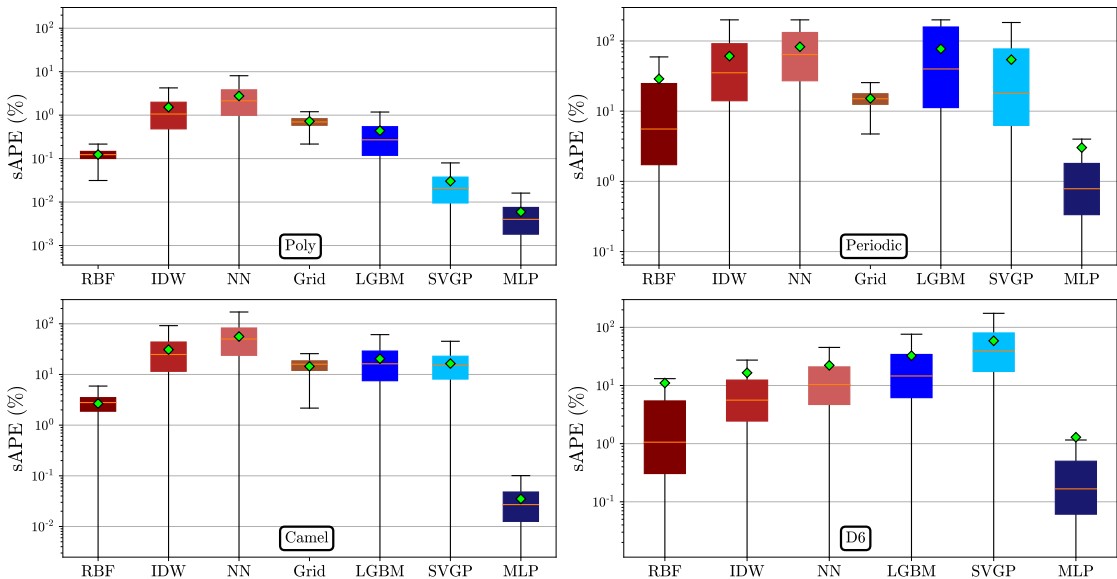

Figure 8: Boxplots of the symmetric absolute percent error (%) for four test functions in 6 dimensions. Red colors indicate interpolation while blue colors indicate regression methods.

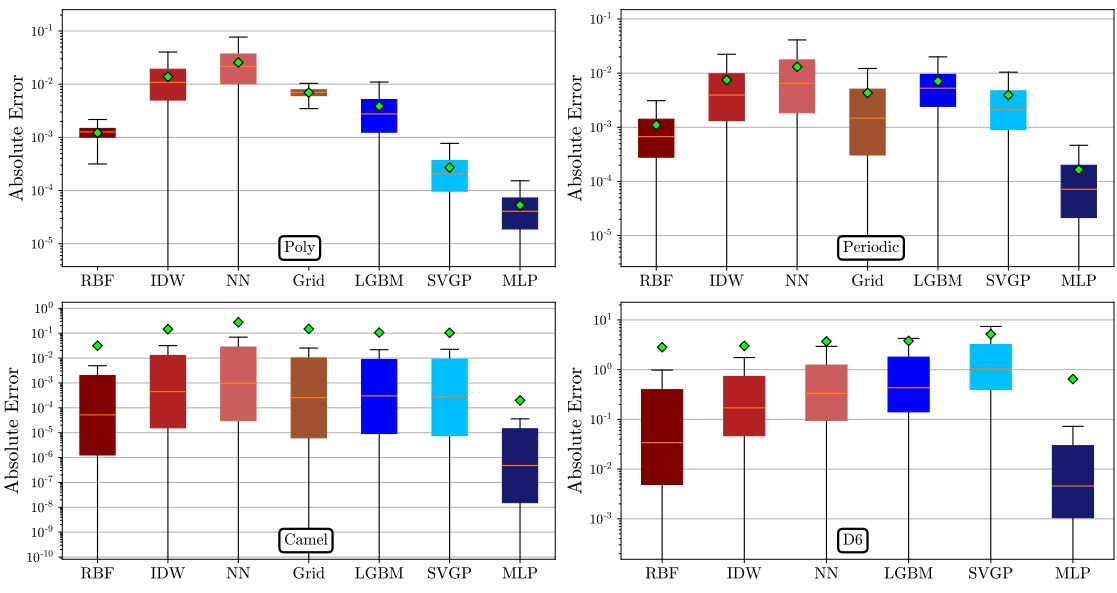

Figure 9: Boxplots of the absolute error for four test functions in 6 dimensions. Red colors indicate interpolation while blue colors indicate regression methods.

Focusing on the results for MLP, the mean sAPE is about 0.1%, which is also about the same as the 3rd quartile, meaning that 75% of the points have sAPEs of 0.1% or less. Considering the speed of evaluation, it took about 19 hours to generate the 5M values of *M* using *TSIL* [61] on a single CPU, whereas the MLP evaluation of 100k testing points takes about 1 second on a CPU (see Fig. 16a), and about 0.3 seconds for 1M points on a GPU. This provides a speed-up of at least 1,400 times on a CPU, and 46,000 times on a GPU.

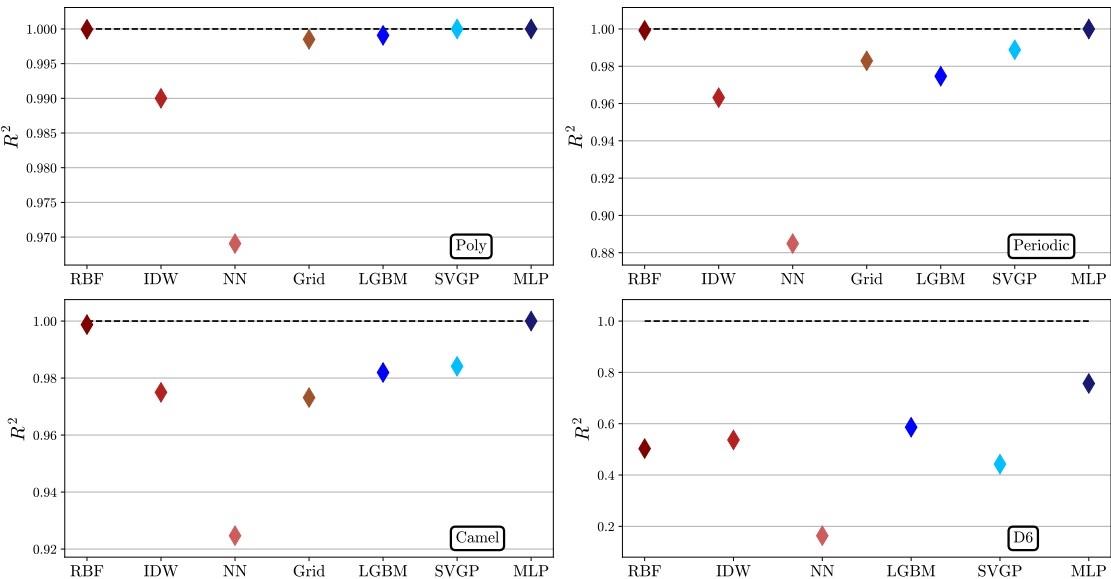

Figure 10: $R^2$ values in 6 dimensions for the three toy functions and D6.

## 6.5 Discussion and Further Analysis

The previous sections showed that for low dimensions, RBF generates good fits and outperforms other approximants in terms of median sAPE for all functions. On the other hand, going to higher dimensions shows the curse of dimensionality taking effect quickly, and not just on RBF, but on other approximants as well, with the exception of MLP. This suggests that MLP is more robust to the curse compared with the other methods. To see this more clearly, we run an experiment on the periodic function where we fix the number of training points while increasing the dimension from 2 to 9. To compare the performance across dimensions, we compute $R^2$ (because it is scale-free), and plot $1 - R^2$ to better show the differences on a log-scale. Figure 15 shows the plot of $1 - R^2$ (0 is best) versus the dimensionality for the periodic function. There is a clear separation between MLP and the other approximants where the curse of dimensionality affects the performance of the other approximants more strongly compared to MLP. Furthermore, the performance of SVGP and LGBM plateaus at dimensions lower than 4, indicating a limitation of these models in achieving higher accuracy for small dimensions.

This is likely due to the scaling properties of these approximants. For RBF, its memory and computational complexities force us to limit the interpolation at an unknown point to 150 nearest neighbors. This does not hinder performance in low dimensions since the density of points is high, but going to larger dimensions, the volume grows exponentially and so does the number of samples required to maintain the same distance between neighboring points. Assuming we sample points on a grid, the distance between adjacent points in 3 dimensions for 5M points is 0.006. To maintain this distance in 9 dimensions, we must sample $10^{20}$ points. Similarly for SVGP, the complexity of fitting a regular GP forces us to utilize a limited number of inducing variables, which become sparse in large dimensions. The same reasoning can be applied to NN and Grid. On the other hand, IDW does not suffer as much from larger amounts of data, but is known to produce a bullseye pattern [68] despite the known points not being maxima, leading to subpar performance. The tendency of LGBM to overfit, even with dart, contributed strongly to its unimpressive performance. As for MLP, the large number of trainable parameters coupled with the nonlinearity at each layer allows for great representational ability. Whereas RBF solves for the weights exactly and thus requiring high computational cost,

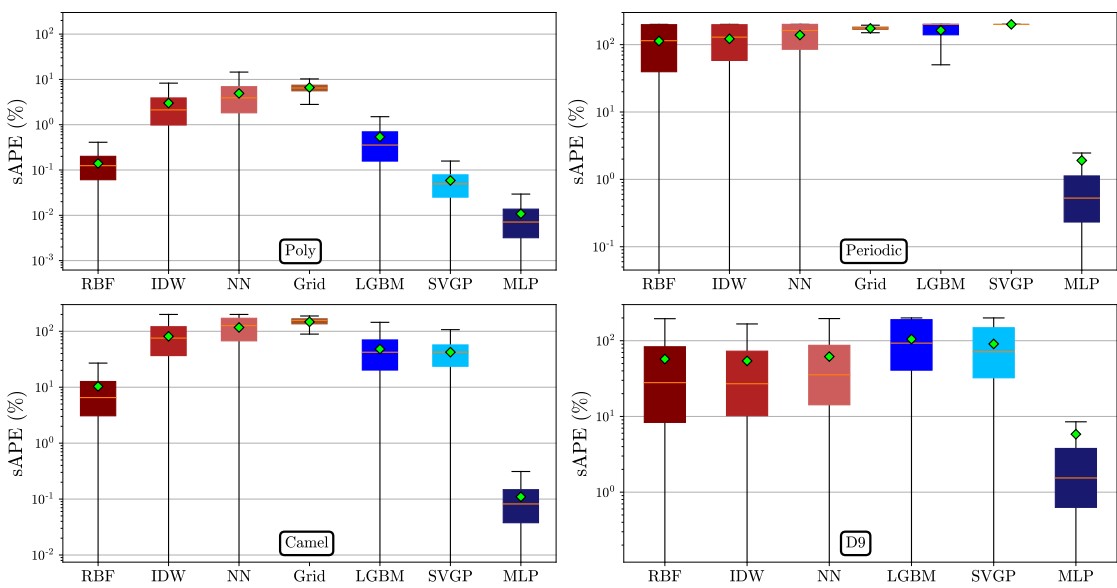

Figure 11: Boxplots of the symmetric absolute percent error (%) for four test functions in 9 dimensions. Red colors indicate interpolation while blue colors indicate regression methods.

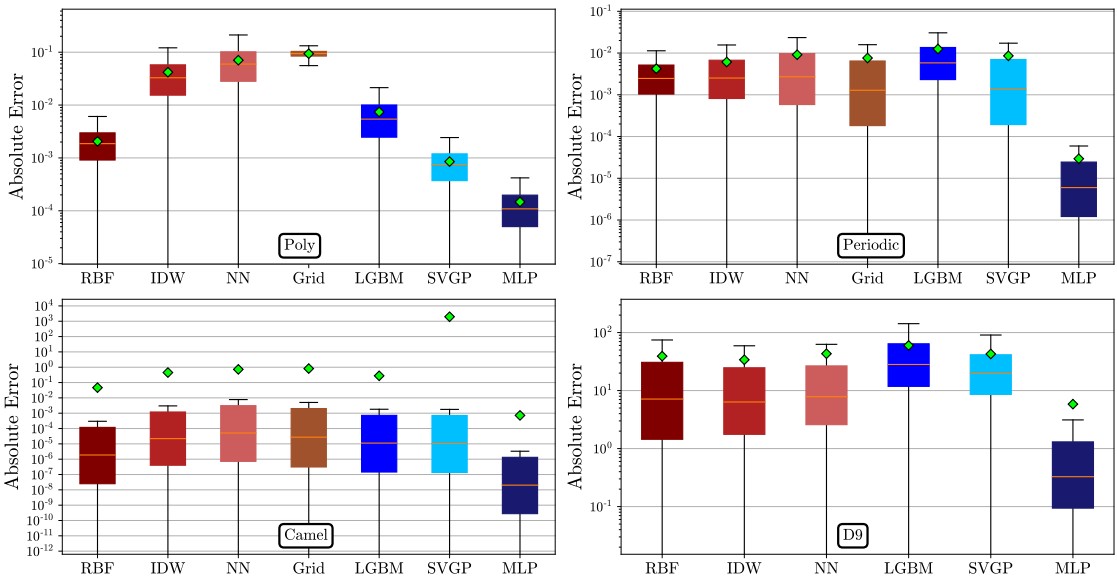

Figure 12: Boxplots of the absolute error for four test functions in 9 dimensions. Red colors indicate interpolation while blue colors indicate regression methods.

adjusting the large number of trainable parameters in the MLP is done in small steps during gradient descent through the efficient backpropagation algorithm. This makes the MLP very flexible in finding a good fit to the data, with the downside of needing longer times to train and the possibility of getting stuck in sub-optimal local minima.

As mentioned in Sec. 5, the evaluation times and model sizes are important for speed and distribution. Figure 16 shows the prediction times (Fig. 16a) in seconds (s) on 100k points and

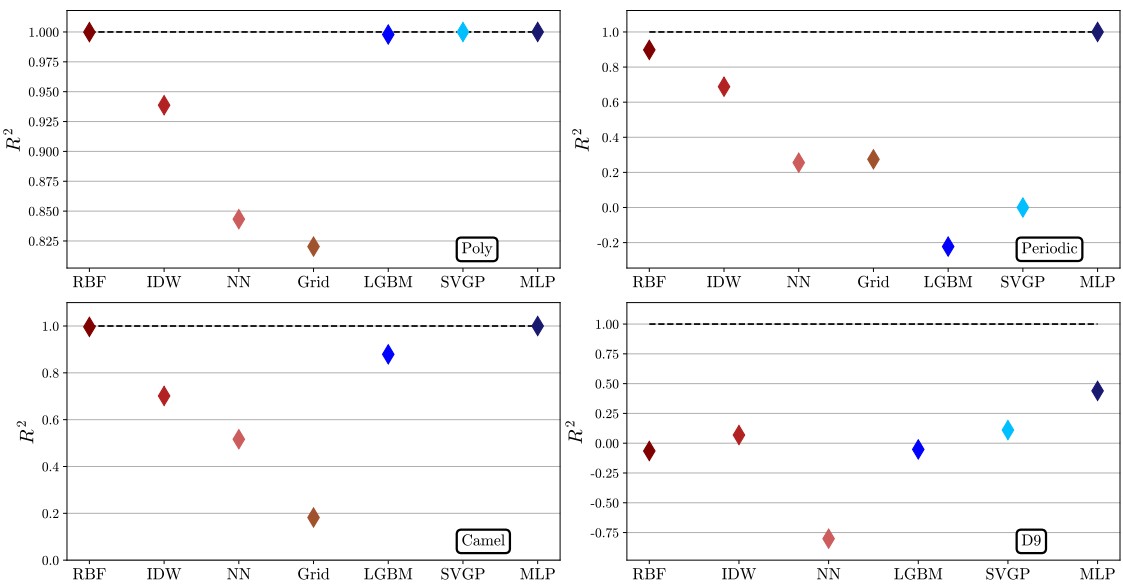

Figure 13: $R^2$ values in 9 dimensions for the three toy functions and D9.

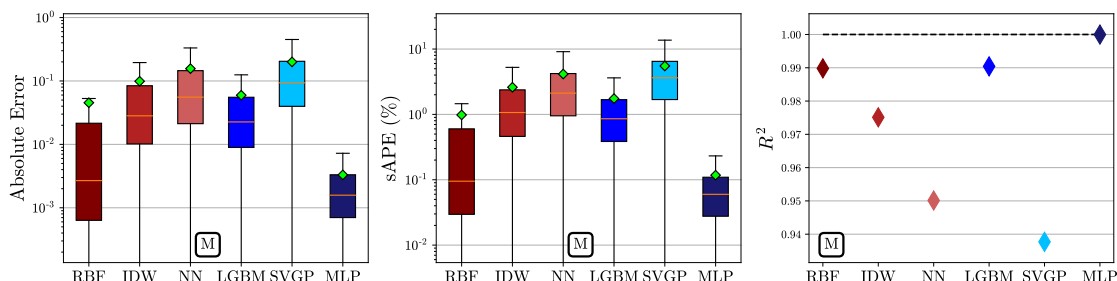

Figure 14: Boxplots of the absolute error (left) and symmetric absolute percent error (middle), and the $R^2$ metric (right) for the two-loop self-energy master integral $M$ in 5 dimensions. Red colors indicate interpolation while blue colors indicate regression methods.

file sizes (Fig. 16b) in megabytes (MB) for the approximants. The darkest shade corresponds to $d = 3$, with lighter shades corresponding to $d = 6$ and $d = 9$. These bar plots show that MLP is fastest in higher dimensions while also requiring the least disk space for storage. On the other hand, training an MLP on large amounts of data can take many hours, whereas the interpolants can fit and predict in a few minutes.

# 7 Conclusions and Future Directions

In this analysis, we studied and compared the performance of interpolation and regression techniques on a variety of functions in increasing dimensionality, two of which are important for precision calculations in high-energy physics. Fixing the number of training points at 5M and varying the dimension from 3 to 9, we find that in low dimensions ($d = 3$) RBF achieves the lowest median AE and sAPE values on all test functions. In higher dimensions ($d = 5$,

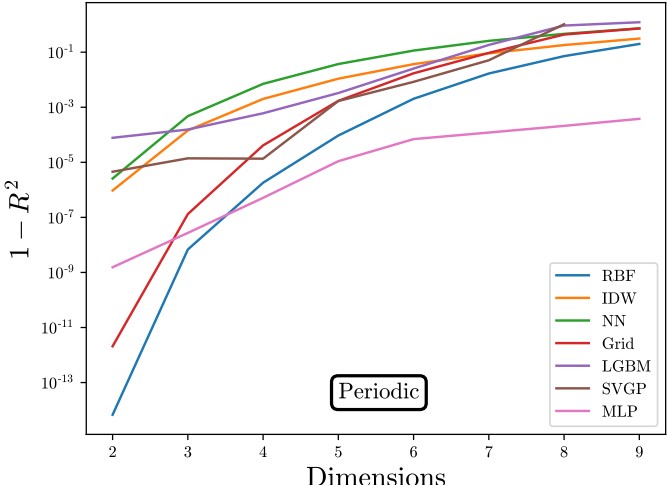

Figure 15: The performance metric $1-R^2$ on the periodic function versus the number of dimensions for all approximants. Lower is better.

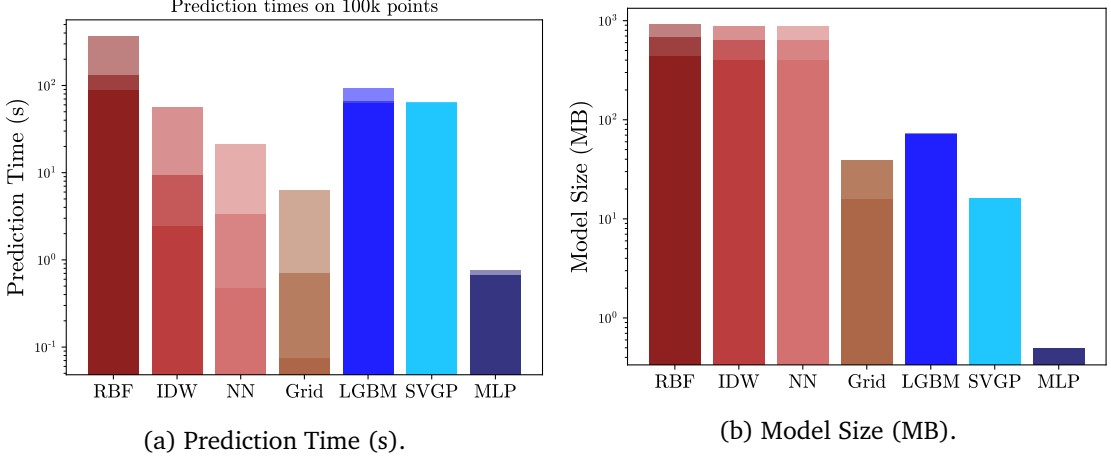

(a) Prediction Time (s).        (b) Model Size (MB).

Figure 16: The prediction times on 100k testing points in seconds (left) and the file size of each approximant in megabytes (right).

$d = 6$ and $d = 9$), we find that MLP outperforms the other approximants, achieving by far the lowest median AE and sAPE values on all test functions. We also find that MLP is fastest at predicting unknown values while having the smallest file size. In addition, MLP is more robust to the curse of dimensionality than other approximants.

There are many interesting avenues to explore moving forward. Although this analysis presents a strong case for MLP in higher dimensions, it is important to investigate the limitations of this method for high-dimensional regression, similar to the analysis in [69] which looked at the statistical information and limitations of generative adversarial networks for event generation in increasing dimensionality. Ultimately, we are interested in how accurate the MLP can potentially be. Future work will look at the dependence of MLP performance on the size of the training set and the model architecture. Furthermore, we would like to investigate various uncertainty estimates such as model ensembling and Monte Carlo Dropout [70]. Answering these questions will shed light on the potential of MLPs and their limitations in accurately approximating computationally expensive functions for high-energy physics.

## Acknowledgments

This research was supported in part through computational resources and services provided by Advanced Research Computing (ARC), a division of Information and Technology Services (ITS) at the University of Michigan, Ann Arbor. This work is supported in part by DOE grant DE-SC0007859.

## A   Relating the Arguments of $D_0$

The Lorentz invariant quantity $s_{23}$ in equation 16 can be related to the other external inputs $s_i$ and a scattering angle $\theta$. Rather than sampling $s_{23}$ directly which has nontrivial limits, we define $x_6 = \frac{\cos\theta + 1}{2}$ and sample $x_6 \in [0, 1]$ instead. The full relation between $u \equiv \frac{s_{23}}{s_{12}}$ and $x_6$ is

$$
\begin{aligned}
u = \frac{1}{2}\Big( &-1 + x_2 + x_3 - x_2 x_3 + x_1(1 + x_3 - x_4) + x_4 + x_2 x_4 + \\
&(2x_6 - 1)\sqrt{x_1^2 + (-1 + x_2)^2 - 2x_1(1 + x_2)}\sqrt{x_3^2 + (-1 + x_4)^2 - 2x_3(1 + x_4)}\Big).
\end{aligned}
\tag{28}
$$

## B   Neighbors for RBF

As mentioned in Sec. 2.4, we must limit the interpolation of RBF to a small number of nearest neighbors. To determine this number, we run experiments on the test functions in various dimensions where we compute the $R^2$ performance of RBF with increasing neighbors. Figure 17 shows the results for various functions in different dimensions. We find that if RBF achieves a high $R^2$ for a function at low neighbors, there are diminishing returns on the performance after about 150 neighbors (Fig. 17, panels 1 – 5). On the other hand, when RBF performs poorly (panel 6, D9 function), it will require a large number of neighbors that brings back the computational and memory overhead we are trying to avoid.

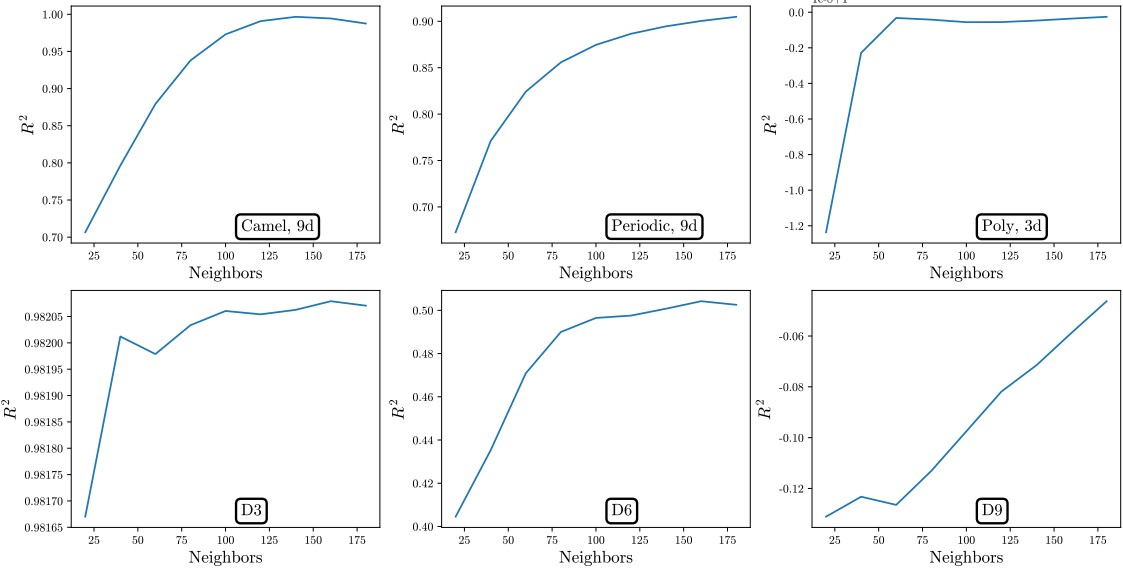

Figure 17: The performance metric $R^2$ of RBF as a function of the nearest neighbors for various functions and dimensions.

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
