# Peer review of "Comparing Machine Learning and Interpolation Methods for Loop-Level Calculations"

_SciPost Physics, doi:SciPost Phys. 12, 187 (2022)_

## Round 2 · Referee Report · Simon Badger (Referee 1) · 2022-1-25

Report

In the article "Function Approximation for High-Energy Physics: Comparing Machine Learning and Interpolation Methods" the authors attempt to address some of the issues associated with the optimisation of complicated functions appearing in current high energy physics simulations.

The article is well prepared and gives a systematic comparison between a variety of modern Machine Learning algorithms and more conventional interpolation methods. To the best of my knowledge the particular comparison presented here has not appeared in the literature before.

My main concern with the analysis is the choice of benchmark test functions and to what extent the conclusions are applicable in a wider context. While the scalar box integral with arbitrary masses is one of the most complicated integrals appearing at NLO, its numerical evaluation is by no means a bottleneck in modern algorithms. The two-loop self-energy is slightly more interesting since the analytic expressions for integrals of this type with arbitrary masses lead to the appearance of elliptic integrals which are difficult to evaluate numerically.

Nevertheless many of the two-loop self energy integrals themselves are by now well understood (in four dimensions) in terms of elliptic polylogarithms which can be efficiently evaluated through q-expansions of iterated Eisenstein integrals (see for example 1912.00077). I suspect that neither of the test integral topologies would present any difficulties for state-of-the-art precision phenomenology.

The fact that these test cases are considered in odd dimensions is strange to me and unmotivated. Applications in high energy physics rarely involve quantum field theories in odd dimensions as far as I am aware, although I would be happy to hear otherwise.

The issue lies in the fact that the best method will clearly depend on the particular problem at hand. It is well known that Machine Learning methods scale better than conventional techniques for high dimensional problems and so the conclusions of the study do not appear to go against conventional wisdom. I am therefore unconvinced whether the additional information obtained by this systematic study would be of use to someone considering the application of such techniques to a more technically demanding computation.

I believe these concerns could be addressed with some suitable clarifications and additional references to alternative approaches to put this work in a wider context. I also have a few specific suggestions that the authors may consider in a revised version.

  1. The introduction summarises some previous attempts to use regression algorithms for high energy physics in the literature. The speed improvement declared in those articles is listed but I believe a suitable explanation of how the times should be compared should be added. The speed up declared refs. [7,8] for the call times of the approximate function is misleading and irrelevant when the generation of the input data dominates the overall evaluation time. The smaller speed up declared in refs [10-12] includes training, testing and interpolation and so represents a realistic measure of the optimisation. The comments related to refs. [10-12] in this article are perfectly well summarised but an explicit statement regarding the comparison with refs. [7,8] would be welcome.

2 The authors may consider referring to the application of Gaussian processes to the approximation of extremely complex two-loop five particle scattering amplitudes in the NNLO computation of pp->3 photons. (1911.00479) and Zaharid/GPTree (https://zenodo.org/record/3571309#.Ye6p_FjMKqA)

  1. Recent work on the evaluation of multi-loop integrals provides useful context for this work (2112.09145). The authors could consider adding a reference.
  • validity: -
  • significance: -
  • originality: -
  • clarity: -
  • formatting: -
  • grammar: -

Author:  Ibrahim Chahrour  on 2022-03-30  [id 2339]

(in reply to Report 1 by Simon Badger on 2022-01-25)

Please see attached file for replies.

Attachment:

Reply_to_Reviewer_1.pdf

---

## Round 2 · Referee Report · Anonymous (Referee 2) · 2022-3-7

Report

The present article studies the general problem of constructing approximations of general functions in the context of high-energy physics, in particular comparing traditional interpolation methods with strategies based on machine learning. The main result of this study is that for the cases they study, which concern some functions that appear in higher order perturbative calculations, traditional interpolation methods work well in low-dimensions but that multi-layer feed forward perceptrons offer the best performance by far in higher dimensions.

The article is well-written, detailed, and presents a systematic study of a general problem with useful findings that should be relevant for HEP practitioners as well as for other scientists working in fields where the same problem arises. I am convinced that it fulfil the quality requirements expected for a scientific publication.

However, I think that before the paper can be considered as suitable for publication the authors should address a number of points, some of them more conceptual and others more practical. I list my comments here, and while I do not expect the authors to fully tackle all of them, I hope that they can be considered as useful suggestions to further improve this nice study.

  • In one respect, the findings of this study are not particularly surprising. It has been known for a long time that multi-layer feed-forward neural networks offer an extremely powerful alternative to traditional interpolation methods, with i.e. the use of neural networks to parametrise deep-inelastic structure functions having been stablished 20 years ago, with subsequent applications becoming a basically mainstream technique in global analyses of non-perturbative QCD quantities from (nuclear) parton distributions to fragmentation functions. Similar techniques have been applied to speed up event generation in Monte Carlo programs as well as to facilitate the evaluation of complex high-dimensional functions that arise in the context of perturbative QCD and electroweak calculation.

  • I would argue that at this point it is the choice of a traditional interpolation algorithm that should be justified, while the use of neural networks are universal unbiased interpolants to parametrise functions in HEP is more or less an off-the-shelf technique. While the authors mention some of these results in their introduction, it may be a bit more accurate to emphasize that by now these are more or less standard techniques in HEP, and that it has been acknowledge that the use of traditional interpolation is restricted to a problems of certain simplicity. In particular, the statement "Although the previous analyses show promise in their respective methods" should be removed: there is a very large body of scientific work that demonstrates that NN-based methods outperform traditional interpolation techniques for all applications except those too simple to present a major bottleneck in any physics study. I believe that stating this in the introduction would reflect better the state of the art in the field. In other works, while the general answer of when it is worth moving to ML-based interpolation methods depends indeed on the problem, it is almost always found that when quality and performance of interpolation starts to limit how far we go into a given problem, it is the point while moving to ML-based tools is the right choice.

  • In this respect, I think the title does not reflect accurately the contents of the paper. The examples the authors consider are not representative of the whole field of HEP but instead are focused on functions that appear in the context of higher-order calculations. So I think it would be appropriate if the authors modified the title to better reflect their focus.

  • I also think that one should mention that in many physics problems that appear in HEP the functions to be parametrised have some physical interpretation, and for instance they are expected to be continuous and smooth. So the general problem of function interpolation in HEP should also address this point. I mention this because the authors try some interpolators such as nearest neighbours which are discontinuous, and hence cannot be applied to some problems of relevance in HEP. For the higher-order QCD functions that the authors study in this study this is probably since since in these case they cannot be associated a physical interpretation, but this is only a subset of the relevant applications of ML-based regression for HEP problems. It would be interesting to consider how results change if some instance one imposes some smoothness criterion on the parametrised functions (my hunch is that if anything this requirement will further strengthen the superior performance of the MLP method). So address this point, would be interesting to add figures of merit such as arc-length to the ones already considered in the paper.

  • Another reason why it should not surprise anyone that ML-based regressors outperform traditional interpolation techniques in high dimensions is that the latter are designed in general for low-dimensional problems, and simply cannot be trusted to work in high dimensions. For example, techniques like chebyshev polynomial interpolation are based on fitting the coefficients of the chebyshev polynomial expansion to the input data, and such fits are quite instable in high-dimensions due to the cancellations between different terms in the expansion. This problem is absent in ML-based methods, where by construction one starts with an smooth function which is then via the training adjusted to describe as well as possible the data.

  • Yes another relevant topic that I am a but surprised the authors do not consider is that of overlearning. In general NN-based regressors are overly flexible and the input data is never "perfect", but will fluctuate around some true value. Given sufficiently large training times, the NN model will learn these fluctuations and thus deviate systematically from this underlying law. Without considering this possible issue in detail it is not possible to draw any solid conclusion about the performance of ML regressors. I understand that for this specific application the data is "perfect", but I would encourage the authors to consider the case where some fluctuations are added to the data, and reassess their analysis in that case.

  • The choice of training algorithm and of the model hyperparameters is also very important in order to assess the performance of ML-based regressors. A poor choice for example of NN architecture may lead to a model under-performing, but the conclusion here is not that the ML regressors is not appropriate but that the method adopted to select the model hyperparameters is non-optimal. Likewise, a wrong choice of training algorithm (or even of family of algorithm, from SGD-like to GA-like techniques) can complicate the interpretation of the results and the benchmarking of the performance of ML regressors as compared to traditional interpolation techniques.

  • The authors state that "performing hyperparameter optimization is computationally expensive so we rely on empirical tests to guide the settings" which in practice means setting the hyperparameters of the model by trial and error. Given that the whole point of this study is to robustly estimate the performance of ML regressors, I believe that the authors should investigate in a bit more systematic way how the choice of model hyperparameters affect the findings of their work. At the very least, the choice of adopted hyperparameters should be better justified; results for different sets of hyperparameters compared; and results using different minimisers (at least for the MLP analysis) benchmarked.

  • For the reasons stated above, I am surprised that they use a "EarlyStopping callback that stops training when no improvement has been made over 400 epochs". This seems to me in general a recipe for overlearning in practical applications (not in toy scenarios of course). The authors should investigate how the presence of noise in their input data (for example coming from MC fluctuations) affects the outcome of their analysis.

  • Why the authors choose "an architecture of 8 hidden layers with 64 nodes each" for the MLP? As pointed out above, a single layer with sufficiently large number of neurons suffices for a general regression task, so the choice of 8 hidden layers seems a bit difficult to justify to me.

  • Concerning the use of Gaussian Processes, it may be appropriate to mention that feed-forward neural networks can be understood as a collection of GPs. So in this respect the techniques discussed in 3.1 and 3.3 are not really independent but rather they are closely related. It is thus a bit weird that the performance of the GP method is so inferior as compared to the MLP, do the authors understand this point?

  • Are the figures of merit evaluated over all the data points or not? What happens if one uses say 80% to construct the interpolation and 20% for the validation? Do the results presented in the paper change a lot? An important benefit of MLP models is that they are reasonably stable upon extrapolation, which is not always the case with traditional interpolation techniques.

  • Another important consideration when choosing an interpolation/regression strategy is related to uncertainty propagation. As mentioned above in most applications one is fitting to data with some fluctuations over the underlying "truth", and it is important to be able to estimate and propagate all uncertainties to the final model. The authors should discuss how the various interpolation/regression strategies considered should be combined with error propagation methods, a key ingredient of realistic applications.

Requested changes

see above

  • validity: good
  • significance: good
  • originality: ok
  • clarity: high
  • formatting: good
  • grammar: excellent

Author:  Ibrahim Chahrour  on 2022-03-30  [id 2340]

(in reply to Report 2 on 2022-03-07)

Please see attached file for replies.

Attachment:

Reply_to_Reviewer_2.pdf

---

## Round 3 · Referee Report · Simon Badger (Referee 1) · 2022-4-5

Report

I was happy to read the responses to both referee reports and am happy with the improvements that have been made to the article.

I can recommend it for publication in its current form.

---

## Round 3 · Referee Report · Anonymous (Referee 2) · 2022-4-19

Report

I am happy to recommend publication of the revised version of this paper.

---

## Round 3 · List of Changes

• Title changed from "Function Approximation for High-Energy Physics: Comparing Machine Learning and Interpolation Methods" to "Comparing Machine Learning and Interpolation Methods for Loop-Level Calculations"
  • Added references [7], [8], [16], [17], [18]
  • Clarified the results of previous papers in the introduction, in particular the gain in speed of references [9] and [10]
  • Clarified the current state of Machine Learning methods in the field of high-energy physics in the introduction

---

## Editorial Decision

published